# A Systematic Review of Nursing Competencies: Addressing the Challenges of Evolving Healthcare Systems and Demographic Changes

**DOI:** 10.3390/nursrep15020056

**Published:** 2025-02-05

**Authors:** Ippolito Notarnicola, Alketa Dervishi, Blerina Duka, Eriola Grosha, Giovanni Gioiello, Sara Carrodano, Gennaro Rocco, Alessandro Stievano

**Affiliations:** 1Department Medicine and Surgical, University of Enna “Kore”, 94100 Enna, Italy; giovanni.gioiello@unikore.it; 2Centre of Excellence for Nursing Scholarship, OPI, 00149 Rome, Italy; sara.carrodano@asl5.liguria.it (S.C.); gennaro.rocco@uniroma2.it (G.R.); alessandro.stievano@unime.it (A.S.); 3Faculty of Technical Medical Sciences, University of Medicine, 1000 Tirana, Albania; alketa.dervishi@umed.edu.al; 4Faculty of Medicine and Surgical, Catholic University “Our Lady of the Good Counsel”, 1000 Tirana, Albania; bleriduka@yahoo.it; 5University of Rome “Tor Vergata”, 00133 Rome, Italy; e.grosha8400@stud.unizkm.al; 6Department of Clinical and Experimental Medicine, University of Messina, 98122 Messina, Italy

**Keywords:** nursing competencies, future, healthcare systems, demographic change, sociocultural phenomena

## Abstract

**Background/Objectives:** The nursing profession is expected to undergo substantial transformations in the coming years due to rapidly evolving sociocultural, environmental, and technological changes. Defining and assessing nursing competencies are essential to ensuring high-quality care and fostering professional development. This systematic review aims to explore the future nursing competencies required and the sociocultural factors influencing their evolution. **Methods:** A comprehensive systematic search was conducted in several indexed databases PubMed, CINAHL, Scopus, and Web of Science databases using keywords such as “nursing competencies”, “future”, “healthcare systems”, and “demographic change”. Studies published within the last five years were included, and a rigorous quality assessment was performed. **Results:** The findings suggest that key sociocultural and technological factors—including environmental sustainability, technological advancements, innovation, globalization, urbanization, policy uncertainty, and demographic shifts—will significantly shape the development of nursing competencies. It is projected that nearly 70% of the nursing workforce will operate in highly unpredictable environments. The review highlights the need to develop interpersonal, higher-order cognitive, and system-level competencies, alongside complementary skills in personal and customer services, decision making, technology utilization, creativity, and scientific methodologies. **Conclusions:** Addressing the future challenges in nursing requires a holistic and strategic approach, including a cultural shift within the profession supported by targeted policies and sustained investment in education and continuous professional development. Training systems should prioritize the development of core competencies and promote lifelong learning to ensure adaptability in an evolving healthcare landscape.

## 1. Introduction

The role of nurses is set to undergo profound changes in the coming years due to rapidly evolving sociocultural, environmental, and technological phenomena. Some authors define competence as “an intrinsic individual characteristic causally linked to effective or superior performance in a task or situation, and which is measured on the basis of a predefined criterion” [1,2].

Nursing competencies can be defined as “a set of complex actions whose result is to guide nursing action in professional practice and which can be evaluated through performance” [3,4]. The definition and assessment of nursing competencies are essential to ensure qualified nursing care for patients and to foster professional development [5].

But what is the future of nursing competencies? To which sociocultural phenomena are they subject? All these sociocultural phenomena, from environmental sustainability to technology, from innovation to globalization, from urbanization to uncertain policies and demographic evolution, will influence the development of nursing competencies in the future [6,7,8].

Estimates suggest that approximately 10% of today’s working nurses are employed in roles whose share of the nursing labor force is projected to grow. To attain universal health coverage worldwide by 2030, an additional nine million nurses will be required. Conversely, there will likely be a decrease of about 20% in the supply of those currently working as nurses. This implies that we are currently unsure about almost 70% of the roles that will be fulfilled by future nurses. [9,10].

But what will happen in the future? Future research on nursing competencies, however, indicates that restructuring roles and retraining nurses may help advance these competencies [9,11]. Numerous professions that are likely to see a decline in employment are probably low- or medium-specialized. Nevertheless, we observe that this is different from other studies: not all jobs with a medium or low level of specialization appear to have the same unavoidable destiny [12].

In general, nurses’ jobs in the public sector, with some exceptions, play a leading role and are expected to grow. The results also indicate that the importance of essential health services is likely to increase. Many of these occupations, again, do not require high specialization. However, they must be combined with a specialist nurse, whom consumers increasingly appreciate [13].

We also expect buoyant demand for some of those professional nursing activities, not all, that reflect the steady growth of health services [14,15]. In this context, a clear indication emerges for education and training systems to focus on a sort of “structural” boundary of people’s talent, the first “skin” within which to strengthen some fundamental competencies and characteristics of the person to manage others as an overall and dynamic capacity, with a view to lifelong learning [16].

It also comes from the necessity of reevaluating “linear” educational systems that run on lengthy cycles and ignore an individual’s basic competencies as well as the vast training programs that disregard the individual and his actual capacity for learning [17,18].

The purpose of this literature review is to examine the future nursing competencies and the sociocultural phenomena that will influence them. It aims to provide a comprehensive understanding of the transformative processes shaping the nursing profession and the key competencies required to navigate the evolving healthcare landscape.

To address the challenges posed by the evolution of healthcare systems and demographic changes, this systematic review was guided by the PICO framework, which clearly defines the population, intervention, comparison, and outcomes of interest in studying the nursing competencies required to tackle these changes [19].Population: Nurses working in healthcare settings characterized by evolving challenges related to demographic changes and transforming healthcare systems.Intervention: Analysis and development of nursing competencies required to address emerging clinical practice needs, including continuous learning, interpersonal, and structural competencies.Comparison: Traditional nursing education models compared to the competencies required by modern and globalized healthcare systems.Outcome: Identification of key competencies to enhance nursing care effectiveness, adaptability to changes, and quality of healthcare delivery in complex settings.

## 2. Materials and Methods

To examine the future nursing competencies and the sociocultural phenomena that will influence them, a comprehensive literature review was conducted.

A comprehensive search was conducted using PubMed, CINAHL, Scopus, and Web of Science databases. Search strings were developed using a combination of key terms and Boolean operators, such as: (“nursing competencies” OR “nursing competence”) AND (“future” OR “demographic change”) AND (“healthcare systems” OR “health care systems”). Filters were applied to limit the search to articles published in English between January 2019 and December 2023, ensuring a focus on the most recent and relevant evidence.

These keywords were selected to capture the key concepts relevant to the study objectives, which were to understand the transformative processes shaping the nursing profession and the core competencies required to navigate the evolving healthcare landscape.

The search was limited to studies published within the last 5 years, from 2019 to 2023. This time frame was chosen to ensure the review focused on the most up-to-date research and projections regarding the future of nursing competencies. Limiting the search to the past 5 years also helped to capture the recent and ongoing impacts of major global events, such as the COVID-19 pandemic, on the nursing profession and the competencies needed.

The search strategy was designed to be broad enough to capture a wide range of relevant literature while also being specific enough to exclude studies that did not directly address the key topics of interest. The combination of keywords allowed the search to identify studies examining nursing competencies from various angles, including the influence of healthcare system changes, demographic shifts, and emerging sociocultural trends.

After the initial database searches, the results were screened for relevance by reviewing the titles and abstracts. Studies that met the inclusion criteria, namely those that focused on the future of nursing competencies and the factors shaping them, were then retrieved in full-text format for a more detailed review and analysis.

The selected studies underwent a rigorous quality assessment to ensure the validity and reliability of the findings. This involved evaluating the study design, methodological approach, and the overall strength of the evidence presented. The quality assessment was guided by established frameworks, such as the Joanna Briggs Institute Critical Appraisal tools, to ensure a standardized and comprehensive evaluation [20]. Only studies that met predetermined quality standards were included in the final synthesis and interpretation of the results.

By conducting a systematic search of the literature, employing a robust methodology, and critically appraising the included studies, this review aims to provide a comprehensive and evidence-based understanding of the future of nursing competencies and the complex sociocultural phenomena that will influence their development.

To ensure a rigorous and relevant selection of studies, specific inclusion and exclusion criteria were applied. Studies were included if they: (1) focused on nursing competencies within the context of evolving healthcare systems and demographic changes; (2) the search was limited to studies published within the last five years, specifically from January 2019 to December 2023, to ensure a focus on the most recent and relevant literature; (3) were written in English; and (4) used quantitative, qualitative, or mixed methods to explore nursing competencies.

Studies were excluded if they: (1) did not directly address nursing competencies or their application in healthcare systems; (2) were commentaries, opinion pieces, or editorials without empirical data; (3) were not available in full text; or (4) focused on unrelated topics such as medical education or non-nursing professions.

These criteria ensured the inclusion of high-quality, relevant studies to support the objectives of this systematic review.

This systematic review was conducted in accordance with the Preferred Reporting Items for Systematic Reviews and Meta-Analyses (PRISMA) 2020 guidelines. The PRISMA checklist was utilized to ensure transparency and rigor in the reporting of the review process [21]. Each stage of the review, including the identification, screening, eligibility assessment, and inclusion of studies, adhered to the PRISMA flow diagram and corresponding checklist items. The checklist ensured that all essential methodological details, such as eligibility criteria, search strategy, data extraction, and quality assessment, were explicitly documented and reported. This approach enhances the reproducibility and reliability of the review findings. The study selection process followed the PRISMA guidelines and was conducted in multiple phases. Initially, the titles and abstracts of the identified articles were independently screened by two reviewers to assess relevance to the study objectives. Full-text articles meeting the inclusion criteria were then further reviewed for final eligibility. Disagreements between reviewers were resolved through discussion or consultation with a third reviewer.

Data were extracted using a predefined matrix that included the following variables: author, year of publication, country, study design, target population, study objectives, interventions considered, key findings, and conclusions. Data quality and consistency were ensured through cross-checking among reviewers.

### Screening and Quality Assessment Process

The screening process for articles involved a multi-stage approach to ensure the inclusion of relevant and high-quality studies. Initially, titles and abstracts were independently reviewed by two authors to assess their alignment with the study objectives. Full-text articles were subsequently evaluated based on predefined inclusion and exclusion criteria. Discrepancies between reviewers were resolved through discussion or consultation with a third author.

The exclusion of studies was based on a qualitative assessment using the Joanna Briggs Institute (JBI) critical appraisal tools. While these tools do not generate a total quality score, studies that did not meet key methodological criteria—such as clear research objectives, appropriate study design, and sufficient data analysis—were excluded to ensure the rigor and reliability of the review findings. This approach was intended to enhance the credibility of the synthesized evidence by focusing on studies that align with the review’s objectives and methodological standards. To minimize the risk of bias and enhance the overall validity of the synthesized evidence, studies that did not meet the predefined quality threshold were excluded. The quality assessment criteria were carefully defined to select studies that contribute to meaningful conclusions and align with the objectives of the review. The use of these tools allowed for a more nuanced understanding of the reliability and validity of the included studies, directly influencing the synthesis of the results and the conclusions drawn.

The collected data were analyzed using a narrative synthesis approach, identifying recurring themes and emerging trends related to future nursing competencies. Findings were grouped into major thematic areas and compared with existing guidelines and nursing practices to identify gaps and opportunities for development.

## 3. Results

The systematic search conducted in the PubMed, CINAHL, Scopus and Web of Science databases, using the keywords “nursing competencies”, “future”, “healthcare systems” and “demographic change”, identified a total of 452 articles published within the last 5 years (Figure 1).

After applying the selection criteria and assessing the methodological quality, 27 studies were included in the final analysis (Table 1).

The findings of the literature review have highlighted several key aspects regarding the future nursing competencies and the sociocultural phenomena that will influence them.

Firstly, it emerged that phenomena such as environmental sustainability, technology, innovation, globalization, urbanization, uncertain policies, and demographic evolution will have a significant impact on the development of nursing competencies [22]. These macrosocial changes will create new challenges and opportunities for the nursing profession, requiring the acquisition of diverse and more complex competencies [23]. For example, nurses will need to be able to address the health consequences of climate change, manage the introduction of new assistive technologies, adapt to migratory flows and transformations in family structures, and navigate a political context characterized by uncertainty and instability.

In terms of projections on the supply and demand of nurses, estimates indicate that approximately 10% of today’s working nurses are employed in areas destined to grow, such as home care, telemedicine services, and elderly care facilities, while there will be a decrease of around 20% in the supply of nurses due to factors such as the retirement of the baby boomer generation and the lack of attractiveness of the profession for younger generations [24]. This implies that the majority of the nursing workforce, almost 70%, will operate in areas characterized by a high degree of uncertainty and unpredictability, such as primary care, emergency services, and specialized hospital wards [25]. This dynamic underscores the need to develop competencies that allow nurses to adapt to rapidly evolving and constantly changing work contexts, such as the ability to address crisis situations, manage the increase in care complexity, and collaborate effectively in multidisciplinary teams.

Future research on nursing competencies suggests that the reorganization of roles and the retraining of nurses can contribute to the development of these competencies [26]. Additionally, it emerged that not all low- or medium-skilled jobs are destined for inevitable decline. On the contrary, some of them, such as nursing roles in the public sector, will play a leading role and are expected to grow [27]. This highlights the importance of investing in the training and continuous professional development of nurses to ensure they are able to adapt and respond effectively to the changing needs of the healthcare system.

A particularly relevant aspect highlighted by the review is the emphasis on the development of interpersonal, higher-order cognitive, and system competencies. Skills such as problem solving, decision making, system analysis and evaluation, and competencies related to a systemic vision will be particularly in demand [28]. These findings underline the need for nurses to acquire competencies that go beyond traditional technical and care-related skills in order to be able to address complex problems, make informed decisions, understand healthcare systems as a whole, and promote continuous quality improvement.

Furthermore, the review has highlighted the importance of complementary competencies such as personal and customer services, decision-making abilities, technological skills, creativity, and the scientific method [29]. These competencies, often considered “soft” skills, will play an increasingly central role in the professional profile of nurses of the future [30]. For example, the ability to establish empathetic relationships with patients and their families, to use digital technologies effectively, to think creatively to address novel problems, and to apply an evidence-based approach will be fundamental to ensuring high-quality nursing care.

Finally, the results underscore the importance of adopting a long-term and comprehensive perspective in the development of the nursing profession in order to redefine nurses’ self-perception, interpersonal interactions, and social role [31]. This will require a cultural shift within the profession, which must be supported by targeted policies and investments in training and continuous professional development. Furthermore, education and training systems will need to focus on the development of “structural” competencies of the person, within a lifelong learning approach, to manage the complexity of future work contexts. This approach will go beyond traditional linear models of education to value individual learning and adaptability capacities.

In summary, the findings of the literature review indicate that the nursing competencies of the future will be strongly influenced by rapidly evolving sociocultural, environmental, and technological phenomena. To address these challenges, nurses will need to develop a broader and more flexible competency profile, encompassing interpersonal skills, higher-order cognitive abilities, and system-level competencies, as well as complementary competencies related to services, technology, and innovation. Moreover, it will be necessary to adopt a holistic and long-term perspective in the development of the nursing profession in order to redefine the social role of nurses and align education and training systems with the future needs of the healthcare system.

The exclusion of articles during the full-text screening was based on specific criteria to ensure the inclusion of studies most relevant to the objectives of this review. The primary reasons for exclusion were categorized as follows:Article excluded (*n* = 76): studies that did not include a sample representative of the target nursing population were excluded to maintain the relevance and applicability of the findings.Not focused on future nursing competencies (*n* = 23): Articles that did not specifically address the future competency requirements for nurses were excluded. These studies primarily focused on current competencies or other unrelated aspects of the nursing profession.Published before the defined time frame (*n* = 16): To ensure that the review captures the most up-to-date literature, studies published before the predefined period of January 2019 to December 2023 were excluded.Irrelevant study design (*n* = 8): Articles with study designs that did not align with the objectives of this review, such as case studies, commentaries, or editorials without empirical data, were excluded.Inaccessible full text (*n* = 4): A few articles were excluded because their full text could not be obtained despite multiple retrieval attempts.

After applying these exclusion criteria, a total of 27 studies were included in the final qualitative synthesis of this literature review (Table 2).

The scores for each criterion reflect the number of studies that met the respective quality standard, out of the total of 27 studies included in the review.

The percentages indicate the proportion of studies that met each criterion.

The total score is calculated out of a maximum of 243 (27 studies × 9 criteria), and the percentage represents the overall quality score.

The overall quality score of 67.1% suggests a reasonably good level of methodological quality in the studies included in the review. The strengths are seen in representative sampling, valid and reliable outcome measurement, appropriate statistical analysis, and consistent interpretation of results. The areas with relatively lower scores are the use of appropriate quantitative and qualitative study designs, as well as the identification and addressing of confounding factors. These represent opportunities for improvement in future research on this topic.

In the Table 3 provides a comprehensive overview of the 27 articles included in the literature review. The studies cover a wide range of critical topics in contemporary healthcare.

These range from assessing advanced airway management by hospital-based paramedics in Saudi Arabia to understanding the variability of the colonic mucosal microbiome in inflammatory bowel diseases associated with arthritis. Additionally, in-depth exploration is conducted on themes such as the correlation between cytological, hormonal, serological, and radiological findings in Hashimoto’s thyroiditis and the protocol requirements for in-home dog food digestibility testing.

The diversity of topics addressed reflects the complexity and breadth of competencies required in the nursing and healthcare contexts today. Furthermore, it is noteworthy to observe the variety of methodologies employed, ranging from systematic reviews to observational studies, highlighting a comprehensive and articulate approach to research in the nursing competencies domain.

This wide array of studies mirrors the myriad challenges and opportunities that nursing professionals must navigate within the rapidly evolving healthcare systems and the evolving needs of patients. Furthermore, it is crucial to acknowledge that the inclusion of seemingly unrelated studies, such as those on dog food digestibility testing protocols, could offer innovative insights for future nursing practice.

These studies may prompt reflection on overlooked yet pertinent aspects, such as the link between animal and human health, thereby paving the way for new avenues of research and nursing competence development in response to emerging challenges in healthcare systems and shifting demographics.

## 4. Discussion

The findings of the present literature review indicate that the nursing competencies of the future will be strongly influenced by a series of rapidly evolving sociocultural, environmental, and technological phenomena. These macrolevel changes will create new challenges and opportunities for the nursing profession, requiring the acquisition of diverse and more complex competencies.

Specifically, nurses will need to develop competencies to address the health consequences of climate change [59], manage the introduction of new assistive technologies [60], adapt to migratory flows and transformations in family structures [32], and navigate a political context characterized by uncertainty and instability [33]. These competencies will go beyond traditional technical and care-related skills, requiring a systemic vision and the ability to address complex problems.

The review included a variety of study types, ranging from quantitative [35,40,41] to qualitative [36,46,61], mixed methods [55], and systematic reviews [57]. These studies highlight the diverse methodologies employed to explore and understand the evolving competencies required in nursing.

The projections on the supply and demand of nurses indicate that the majority of the nursing workforce, almost 70%, will operate in areas characterized by a high degree of uncertainty and unpredictability. This underscores the need to develop adaptation and resilience competencies to enable nurses to respond effectively to the changing needs of the healthcare system [53]. Studies such as those by Greeff et al. (2020) and Weber et al. (2022) emphasize the impact of the COVID-19 pandemic on nursing competencies and the critical need for training in resilience and adaptability [44,61].

Despite these challenges, future research suggests that the reorganization of roles and the retraining of nurses can contribute to the development of the necessary competencies. For instance, Alenazi et al. (2021) and Arrogante et al. (2021) discuss the importance of advanced airway management and cardiac arrest management in enhancing nursing competencies through simulation-based assessments [35,55]. Moreover, roles in the public sector, such as those explored by Phalatse et al. (2022), are destined to grow, highlighting the importance of investing in continuous training [45].

A key aspect that emerged from the review is the emphasis on the development of interpersonal, higher-order cognitive, and system-level competencies. These skills, which go beyond traditional technical competencies, will enable nurses to address complex problems, make informed decisions, understand healthcare systems as a whole, and promote continuous quality improvement. This is supported by studies such as those by Dhuper et al. (2022) and Emsley et al. (2022), which focus on trauma-informed care and support programs for healthcare providers [34,55].

Furthermore, complementary competencies such as personal and customer services, decision-making abilities, technological skills, creativity, and scientific methods will play an increasingly central role in the professional profile of nurses of the future. Studies by DeGrande et al. (2022) and Gresham et al. (2020) underscore the importance of these “soft” competencies in ensuring high-quality nursing care [37,59].

In conclusion, to address future challenges, it will be necessary to adopt a long-term and comprehensive perspective in the development of the nursing profession. This will require a cultural shift within the profession, supported by targeted policies and investments in training and continuous professional development. Additionally, education and training systems will need to focus on the development of “structural” competencies of the person, within a lifelong learning approach, to manage the complexity of future work contexts. This approach will go beyond traditional linear models of education to value individual learning and adaptability capacities.

In summary, the present literature review underscores the importance of a holistic and strategic vision in the development of the nursing competencies of the future. Only through cultural and organizational change, supported by adequate investments in training, will nurses be able to acquire the necessary competency profile to address emerging challenges and ensure high-quality healthcare.

### 4.1. Limitations

Despite the findings of the present literature review providing important insights into the future nursing competencies and the sociocultural phenomena that will influence them, the study presents some limitations to be considered.

First, the search was limited to articles published within the last 5 years, in order to focus on the most recent trends and projections. However, this approach may have excluded earlier studies that could have provided additional insights into long-term nursing competencies.

Additionally, the review was based solely on academic articles published in peer-reviewed journals. Although this approach ensures the quality and reliability of the sources, it is possible that other types of documents, such as government reports or expert analyses, which could have enriched the understanding of the phenomenon, were omitted.

Another limitation concerns the selection and analysis of the studies included. Despite the application of rigorous quality criteria, the review remains subjective to the interpretation of the researchers. Potential biases or preconceptions could have influenced the selection and interpretation of the results.

Finally, the present review focused primarily on the global perspective of future nursing competencies. However, there may be regional differences or specific contexts that were not sufficiently explored, which could require further investigation.

Despite these limitations, the review provides a comprehensive and up-to-date analysis of the emerging trends regarding the future nursing competencies, offering valuable insights for research, education, and the development of the nursing profession.

### 4.2. Implications for Future Research

The findings of this systematic review highlight several opportunities for future research. First, there is a need to develop standardized tools for assessing nursing competencies across diverse healthcare settings, ensuring their applicability in both high-resource and low-resource environments. Additionally, longitudinal studies could explore how the implementation of targeted educational programs impacts the development and application of these competencies in practice.

Future research should also investigate the relationship between nursing competencies and patient outcomes, particularly in the context of evolving healthcare challenges such as aging populations, technological advancements, and global health crises. Moreover, qualitative studies could provide deeper insights into nurses’ perceptions of competency requirements and the barriers they face in achieving them.

Finally, cross-cultural and comparative studies would be valuable in understanding how nursing competencies are shaped by sociocultural, economic, and healthcare system differences. Such studies could guide the adaptation of global competency frameworks to local contexts, ensuring their relevance and effectiveness in addressing the unique challenges of each healthcare system.

The findings of this review highlight the critical need for a seamless integration between pre-licensure nursing education and continuing education to ensure the development and reinforcement of essential competencies throughout a nurse’s career. Pre-licensure programs, such as those guided by frameworks like the AACN Essentials, play a foundational role in equipping future nurses with core competencies that align with evolving healthcare needs. However, as the healthcare landscape continues to advance, it becomes imperative that continuing education builds upon these foundational skills, ensuring that nurses remain adaptable and competent in addressing emerging challenges. Collaboration between academic institutions and healthcare organizations is essential to create a structured pathway for lifelong learning, fostering an environment where theoretical knowledge and practical competencies coalesce to enhance patient care and professional development. Future initiatives should focus on developing competency frameworks that bridge pre-licensure and continuing education, ensuring consistency and alignment with the dynamic demands of healthcare practice.

## 5. Conclusions

The present literature review has provided a comprehensive analysis of the future nursing competencies and the key sociocultural phenomena that will shape their development. The findings underscore the importance of adopting a holistic and strategic approach to address the challenges and opportunities facing the nursing profession in the years to come.

The review has highlighted the need for nurses to acquire a diverse and flexible set of competencies, going beyond traditional technical and care-related skills. Specifically, the emphasis on the development of interpersonal, higher-order cognitive, and system-level competencies reflects the increasing complexity of healthcare contexts and the need for nurses to possess a systemic understanding of the evolving healthcare landscape.

Moreover, the review has identified the growing significance of complementary competencies, such as personal and customer services, decision-making abilities, technological skills, creativity, and the scientific method. These “soft” competencies play a critical role in ensuring high-quality, patient-centered nursing care in the future.

To support the development of these future-oriented nursing competencies, the findings underline the necessity for a cultural shift within the profession, accompanied by targeted policies and investments in training and continuous professional development. Education and training systems will also need to adapt, moving beyond linear models to focus on the cultivation of “structural” competencies and lifelong learning capacities. This approach emphasizes the need for resilience and adaptability in nursing education.

In conclusion, the present review highlights the critical importance of proactively addressing the evolving competency requirements of the nursing profession. By adopting a long-term, holistic perspective and implementing strategic initiatives, the nursing community can ensure that the profession is equipped to meet the complex healthcare needs of the future and continue to deliver exceptional care to patients and communities worldwide.

## Figures and Tables

**Figure 1 nursrep-15-00056-f001:**
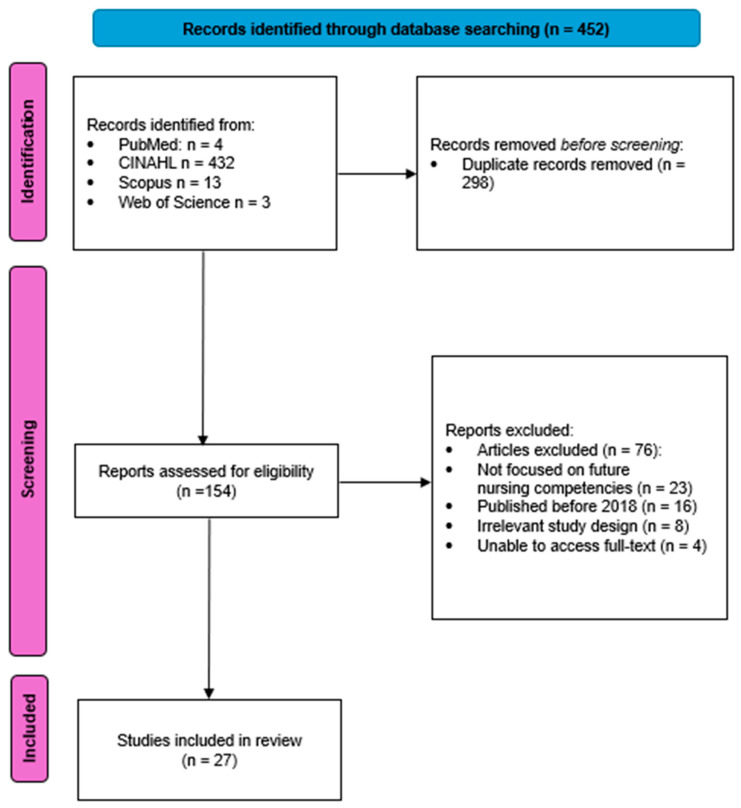
PRISMA statement.

**Table 1 nursrep-15-00056-t001:** Search strategy.

Search Strategy		
Database	String	Record
PubMed	(“nursing competencies” [Title/Abstract] OR “nursing competence” [Title/Abstract]) AND (“future” [Title/Abstract] OR “demographic change” [Title/Abstract]) AND (“healthcare systems” [Title/Abstract] OR “health care systems” [Title/Abstract])	4
CINAHL	(MH “Nursing Competence”) AND (TI “future” OR TI “demographic change”) AND (TI “healthcare systems” OR TI “health care systems”)	432
Scopus	TITLE-ABS-KEY (“nursing competencies” OR “nursing competence”) AND TITLE-ABS-KEY (future OR “demographic change”) AND TITLE-ABS-KEY (“health care systems” OR “healthcare systems”)	13
Web of Science	TS = (“nursing competencies” OR “nursing competence”) AND TS = (future OR “demographic change”) AND TS = (“health care systems” OR “healthcare systems”)	3
Total		452

**Table 2 nursrep-15-00056-t002:** JBI Quality Appraisal Criteria Score.

Category	Criteria	Count	Percentage	Articles
Study design	Appropriate quantitative studies	dic-27	44.4%	Alenazi et al., 2021; Alizadeh et al., 2024; Arrogante et al., 2021; Bos et al., 2023; Botha et al., 2020; Drummer et al., 2021; Gresham et al., 2020; Ng et al., 2022; Nousiainen et al., 2020; Ong et al., 2019; Park et al., 2022; Wickwire et al., 2022
	Appropriate qualitative studies	ago-27	29.6%	Arrogante et al., 2021; DeGrande et al., 2022; Dhuper et al., 2022; Emsley et al., 2022; Greeff et al., 2020; Phalatse et al., 2022; Vashisht et al., 2023; Weber et al., 2022
	Total	20/27	74.0%	
Sampling	Representative sampling	22/27	81.5%	Alenazi et al., 2021; Alizadeh et al., 2024; Almahari et al., 2023; Arrogante et al., 2021; Arrogante et al., 2021; Botha et al., 2020; DeGrande et al., 2022; Dhuper et al., 2022; Drummer et al., 2021; Emsley et al., 2022; Gresham et al., 2020; Greeff et al., 2020; Ng et al., 2022; Nousiainen et al., 2020; Ong et al., 2019; Park et al., 2022; Peiró et al., 2020; Phalatse et al., 2022; Pires et al., 2020; Stecher et al., 2020; Vashisht et al., 2023; Weber et al., 2022
	Non-representative sampling	mag-27	18.5%	Bos et al., 2023; Danielis et al., 2019; Domingues Sousa et al., 2022; Hartnack and Roos, 2021; Wickwire et al., 2022
Confounding factors	Identified and addressed adequately	20/27	74.1%	Alenazi et al., 2021; Alizadeh et al., 2024; Almahari et al., 2023; Arrogante et al., 2021; Arrogante et al., 2021; Botha et al., 2020; DeGrande et al., 2022; Dhuper et al., 2022; Drummer et al., 2021; Emsley et al., 2022; Gresham et al., 2020; Greeff et al., 2020; Ng et al., 2022; Nousiainen et al., 2020; Ong et al., 2019; Park et al., 2022; Peiró et al., 2020; Phalatse et al., 2022; Pires et al., 2020; Vashisht et al., 2023
	Not identified/addressed adequately	lug-27	25.9%	Bos et al., 2023; Danielis et al., 2019; Domingues Sousa et al., 2022; Hartnack and Roos, 2021; Stecher et al., 2020; Weber et al., 2022; Wickwire et al., 2022
Outcome measurement	Valid and reliable measurement	24/27	88.9%	Alenazi et al., 2021; Alizadeh et al., 2024; Almahari et al., 2023; Arrogante et al., 2021; Arrogante et al., 2021; Bos et al., 2023; Botha et al., 2020; Dhuper et al., 2022; Drummer et al., 2021; Emsley et al., 2022; Gresham et al., 2020; Greeff et al., 2020; Ng et al., 2022; Nousiainen et al., 2020; Ong et al., 2019; Park et al., 2022; Peiró et al., 2020; Phalatse et al., 2022; Pires et al., 2020; Stecher et al., 2020; Vashisht et al., 2023; Weber et al., 2022; Wickwire et al., 2022
	Invalid/unreliable measurement	mar-27	11.1%	Danielis et al., 2019; DeGrande et al., 2022; Domingues Sousa et al., 2022
Statistical analysis	Appropriate analysis	24/27	88.9%	Alenazi et al., 2021; Alizadeh et al., 2024; Almahari et al., 2023; Arrogante et al., 2021; Arrogante et al., 2021; Bos et al., 2023; Botha et al., 2020; Dhuper et al., 2022; Drummer et al., 2021; Emsley et al., 2022; Gresham et al., 2020; Greeff et al., 2020; Ng et al., 2022; Nousiainen et al., 2020; Ong et al., 2019; Park et al., 2022; Peiró et al., 2020; Phalatse et al., 2022; Pires et al., 2020; Stecher et al., 2020; Vashisht et al., 2023; Weber et al., 2022; Wickwire et al., 2022
	Inappropriate analysis	mar-27	11.1%	Danielis et al., 2019; DeGrande et al., 2022; Domingues Sousa et al., 2022
Interpretation of results	Interpretation consistent with results	25/27	92.6%	Alenazi et al., 2021; Alizadeh et al., 2024; Almahari et al., 2023; Arrogante et al., 2021; Arrogante et al., 2021; Bos et al., 2023; Botha et al., 2020; Dhuper et al., 2022; Drummer et al., 2021; Emsley et al., 2022; Gresham et al., 2020; Greeff et al., 2020; Ng et al., 2022; Nousiainen et al., 2020; Ong et al., 2019; Park et al., 2022; Peiró et al., 2020; Phalatse et al., 2022; Pires et al., 2020; Stecher et al., 2020; Vashisht et al., 2023; Weber et al., 2022; Wickwire et al., 2022; Hartnack and Roos, 2021; Greeff et al., 2020
	Interpretation not consistent	feb-27	7.4%	Danielis et al., 2019; Domingues Sousa et al., 2022
Total score		163/243	67.1%	

**Table 3 nursrep-15-00056-t003:** Study included in review.

N	Authors	Title	Sample	Main Topic	Study Type
1	Alenazi et al., 2021 [32]	Perception and success rate of using advanced airway management by hospital-based paramedics in the Kingdom of Saudi Arabia	Hospital-based paramedics	Advanced airway management	Quantitative
2	Alizadeh et al., 2024 [33]	The colonic mucosal microbiome in ibd-associated arthritis is more variable than in ibd alone	IBD patients with and without arthritis	Microbiome variability	Quantitative
3	Almahari et al., 2023 [34]	Hashimoto Thyroiditis beyond Cytology: A Correlation between Cytological, Hormonal, Serological, and Radiological Findings	Patients with Hashimoto Thyroiditis	Multi-modal diagnostics	Correlational study
4	Arrogante et al., 2021 [35]	Comparing formative and summative simulation-based assessment in undergraduate nursing students	Nursing students	Simulation-based assessment	Quantitative
5	Arrogante et al., 2021 [36]	Reversible causes of cardiac arrest: Nursing competency acquisition and clinical simulation satisfaction in undergraduate nursing students	Nursing students	Cardiac arrest management	Mixed methods
6	Bos et al., 2023 [37]	Determining the protocol requirements of in-home dog food digestibility testing	Dogs	Digestibility testing protocols	Quantitative
7	Botha et al., 2020 [38]	American College of Radiology Thyroid Imaging Reporting and Data System standardises reporting of thyroid ultrasounds	Patients undergoing thyroid ultrasound	Standardized reporting	Quantitative
8	Danielis et al., 2019 [39]	Nursing Sensitive Outcomes in the Intensive Care Unit: a scoping review protocol	ICU patients	Nursing-sensitive outcomes	Scoping review protocol
9	DeGrande et al., 2022 [40]	Aesthetics and Technology Integration in a Community-Based Primary Care Nursing Curriculum	Nursing students and educators	Curriculum development	Qualitative
10	Dhuper et al., 2022 [41]	Future public health emergencies and disasters: sustainability and insights into support programs for healthcare providers	Healthcare providers	Support programs	Qualitative
11	Domingues Sousa et al., 2022 [42]	Estudo de caso de um adolescente segundo o Modelo Teórico do Ajuste ao Cancro Parental	Adolescent with parental cancer	Adolescent adjustment	Case study
12	Drummer et al., 2021 [43]	Hyperlipidemia May Synergize with Hypomethylation in Establishing Trained Immunity and Promoting Inflammation in NASH and NAFLD	Patients with NASH and NAFLD	Inflammation and trained immunity	Quantitative
13	Emsley et al., 2022 [44]	Trauma-informed care in the UK: where are we?	Health policies and professionals	Trauma-informed care	Qualitative
14	Gresham et al., 2020 [45]	Evaluating Treatment Tolerability in Cancer Clinical Trials Using the Toxicity Index	Cancer patients in clinical trials	Treatment tolerability	Quantitative
15	Greeff et al., 2020 [46]	Exploring the impact of COVID-19 on nursing competency and training	Nurses during the COVID-19 pandemic	Competency and training impact	Qualitative
16	Hartnack and Roos, 2021 [47]	Teaching: confidence, prediction and tolerance intervals in scientific practice: a tutorial on binary variables	Tutorial participants	Statistical methods	Tutorial
17	Ng et al., 2022 [48]	Use of Web-Based Calculator for the Implementation of ACR TI-RADS Risk-Stratification System	Patients undergoing thyroid ultrasound	Risk stratification	Quantitative
18	Nousiainen et al., 2020 [49]	Retention of metals in periprosthetic tissues of patients with metal-on-metal total hip arthroplasty	Patients with hip arthroplasty	Metal retention	Quantitative
19	Ong et al., 2019 [50]	Do current Philips ultrasound systems exceed the recommended safety limits during routine prenatal ultrasounds?	Prenatal ultrasound patients	Ultrasound safety	Quantitative
20	Park et al., 2022 [51]	Osteogenic Differentiation of Human Mesenchymal Stem Cells Modulated by Surface Manganese Chemistry in SLA Titanium Implants	Human mesenchymal stem cells	Osteogenic differentiation	Quantitative
21	Peiró et al., 2020 [52]	The COVID-19 crisis: Skills that are paramount to build into nursing programs for future global health crisis	Nursing students and educators	Nursing skills	Narrative review
22	Phalatse et al., 2022 [53]	Occupational therapists’ perspectives on the impact of COVID-19 lockdowns on their clients in Gauteng, South Africa	Occupational therapists and clients	Impact of lockdowns	Qualitative
23	Pires et al., 2020 [54]	Wettability and pre-osteoblastic behavior evaluations of a dense bovine hydroxyapatite ceramics	Pre-osteoblastic cells	Biomaterials	Quantitative
24	Stecher et al., 2020 [55]	Systematic Review and Meta-analysis of Treatment Interruptions in Human Immunodeficiency Virus (HIV) Type 1–infected Patients Receiving Antiretroviral Therapy	HIV patients	Treatment interruptions	Systematic review and meta-analysis
25	Vashisht et al., 2023 [56]	Challenges Faced and Coping Strategies Adopted by Injecting Drug Users during COVID-19 Lockdown	Injecting drug users	COVID-19 lockdown impact	Qualitative
26	Weber et al., 2022 [57]	The Essence and Role of Nurses in the Future of Biomedical and Health Informatics	Nurses and informatics professionals	Health informatics	Narrative review
27	Wickwire et al., 2022 [58]	Cardiac events and economic burden among patients with hypertension and treated insomnia in the USA	Patients with hypertension and insomnia	Cardiac events and economic burden	Quantitative

## Data Availability

The data presented in this study is available on request from the corresponding author.

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
