# Peer review of "A Systematic Review of Nursing Competencies: Addressing the Challenges of Evolving Healthcare Systems and Demographic Changes"

_nursrep, 2025, doi:10.3390/nursrep15020056_

Round 1
Reviewer 1 Report
Comments and Suggestions for Authors
Well done review on important and needed area for continuing to push nursing forward and prepare for next trends. Recommendations below are intended to tighten areas and enhance reader uptake on a strong article.
Page 1/line 27 (Abstract): Consider naming the specific approach/quality indicators in place of “underwent rigorous quality assessment” to help readers seeking specific review methodology see this in abstract and add to weight of review done. If need word count room to do so, would not lose importance of information to summarize at line 25 as “conducted in several indexed databases” since this will be detailed in article and is less key to detail here than method of assessment.
Pg 2/line 46 and 50: Would remove the lead ins (According to… and As for…) from these term definitions – with the references at end, sufficient and smoother to overqualify it.
Pg 2/line 60: recommend adding a descriptor to first and last sentences that clarify meaning as being nursing roles/specific concentrations (such as … in a role whose share of the nursing labor force is projected to grow.” In last sentence, for further clarity, perhaps: “This implies we are currently unsure about almost 70% of the roles that will be fulfilled by future nurses”
Pg 3/line 95: recommend adding a core reference for the use of the PICO framework
Pg 3/line 118-122: time range is 6 years?
Pg 3/line 133-178: add reference for JB critical appraisal and PRISMA tools. Would recommend being more direct (not “tools such as”, but saying “using X and Y”), referencing the source of the methods.
Pg 7/line 258: recommend describing/defining all the exclusion areas to be transparent (i.e. on next page, the listing of numbers excluded for other reasons such as non-representative sampling would be better with outline on this page defining all exclusion types)
Discussion: consider adding a brief consideration of how pre-licensure nursing education and continuing education will need to collaborate to introduce in pre-licensure and then build upon competencies in continuing education setting. Could be a good highlight to include the coalescing of practice competencies with the pre-licensure competencies such as the AACN Essentials.
Editorial changes recommended:
Pg 3/line 111: three should be four?
Check paper throughout and correct for “competence vs competency/competencies” (examples on Lines 298, 319)
Throughout paper, review how specific roles are named to consider how to make this clearer as being roles within nursing vs. different jobs/professions. Ex: line 63/64 above; line 69 (would be clearer to say “numerous nursing roles” instead of professions, as professions are usually “whole” professions such as nursing, medicine, social worker). Ensure full meaning of sentence can be understood/not dependent on sentence prior for the description of roles/jobs to be understood as nursing (ex: line 218, adding “nursing” between medium-skilled and jobs would more clearly focus sentence).
Throughout paper, screen for the use of the term “that” and remove where not needed for sentence structure, to enhance smooth flow for reader (examples of where it can be removed: line 31, 209, 249, 314, 331)
Author Response
Comment 1: Page 1/line 27 (Abstract): Consider naming the specific approach/quality indicators in place of “underwent rigorous quality assessment” to help readers seeking specific review methodology see this in abstract and add to weight of review done. If need word count room to do so, would not lose importance of information to summarize at line 25 as “conducted in several indexed databases” since this will be detailed in article and is less key to detail here than method of assessment.
Response 1: Thank you for your valuable feedback on the abstract. We appreciate your suggestion to specify the quality assessment approach used in place of the general phrase “underwent rigorous quality assessment” to enhance clarity and provide additional weight to our review methodology. In response to your recommendation, we have revised the abstract to explicitly mention the use of the Joanna Briggs Institute (JBI) critical appraisal tools, which were applied to evaluate the methodological quality of the included studies. This revision provides readers with a clearer understanding of the quality assessment process. Additionally, as suggested, we have adjusted the wording to summarize the database search more concisely by stating “conducted in several indexed databases,” allowing us to allocate space for the more critical methodological details.
Comment 2: Pg 2/line 46 and 50: Would remove the lead ins (According to… and As for…) from these term definitions – with the references at end, sufficient and smoother to overqualify it.
Response 2: Thank you for your helpful suggestion regarding the phrasing in lines 46 and 50. We agree that the lead-in phrases ("According to..." and "As for...") may overqualify the definitions and that removing them will improve the readability and flow of the text. In response to your comment, we will revise these sentences by directly presenting the definitions, ensuring that the references at the end provide sufficient attribution without unnecessary introductory phrases. This adjustment will enhance the clarity and conciseness of the manuscript.
Comment 3: Pg 2/line 60: recommend adding a descriptor to first and last sentences that clarify meaning as being nursing roles/specific concentrations (such as … in a role whose share of the nursing labor force is projected to grow.” In last sentence, for further clarity, perhaps: “This implies we are currently unsure about almost 70% of the roles that will be fulfilled by future nurses”
Response 3: Thank you for your insightful suggestion regarding the clarification of meaning in line 60. We acknowledge the importance of specifying the context of nursing roles to ensure clarity for readers. In response to your comment, we will revise the first sentence to include a descriptor, such as "... in a role whose share of the nursing labor force is projected to grow," to better convey the focus on nursing-specific concentrations. Additionally, we will update the last sentence for improved clarity by stating, “This implies we are currently unsure about almost 70% of the roles that will be fulfilled by future nurses,” as recommended.
Comment 4: Pg 3/line 95: recommend adding a core reference for the use of the PICO framework
Response 4: Thank you for your valuable suggestion regarding the inclusion of a core reference for the use of the PICO framework in line 95. We acknowledge the importance of providing a well-established citation to support the methodological approach used in our study. In response to your comment, we will incorporate an appropriate reference to a widely recognized source that details the application and relevance of the PICO framework in systematic reviews. This addition will enhance the credibility and methodological rigor of our study.
Comment 5: Pg 3/line 118-122: time range is 6 years?
Response 5: Thank you for pointing out the inconsistency in the reported time range on lines 118-122. We acknowledge the oversight and appreciate the opportunity to clarify this issue. In line with our previous response regarding the search period, the correct time frame for the included studies is from January 2019 to December 2023, covering a period of five years. We will revise the manuscript to ensure consistency throughout and to accurately reflect the intended study period.
Comment 6: Pg 3/line 133-178: add reference for JB critical appraisal and PRISMA tools. Would recommend being more direct (not “tools such as”, but saying “using X and Y”), referencing the source of the methods.
Response 6: Thank you for your valuable feedback regarding the need for explicit references to the Joanna Briggs Institute (JBI) critical appraisal tools and the PRISMA guidelines in lines 133-178. In response to your comment, we have revised the text to explicitly state that the quality assessment was conducted using the Joanna Briggs Institute (JBI) critical appraisal tools and that the reporting adhered to the PRISMA 2020 guidelines, providing appropriate citations for both. These revisions ensure clarity and transparency in our methodological approach and directly address your recommendation to avoid ambiguous phrasing such as “tools such as.”
Comment 7: Pg 7/line 258: recommend describing/defining all the exclusion areas to be transparent (i.e. on next page, the listing of numbers excluded for other reasons such as non-representative sampling would be better with outline on this page defining all exclusion types)
Response 7: Thank you for your insightful suggestion regarding the need to describe and define all exclusion criteria more transparently in line 258. In response to your comment, we will revise the manuscript to provide a clearer and more comprehensive outline of the exclusion criteria within this section. Specifically, we will detail the various reasons for exclusion, such as non-representative sampling, lack of methodological rigor, and irrelevance to the study objectives. This clarification will ensure that readers have a complete understanding of the selection process and improve the overall transparency of the review. Additionally, we will ensure consistency by aligning this information with the exclusion numbers presented later in the manuscript, allowing for a more structured and coherent presentation of the study selection process.
Comment 7: Discussion: consider adding a brief consideration of how pre-licensure nursing education and continuing education will need to collaborate to introduce in pre-licensure and then build upon competencies in continuing education setting. Could be a good highlight to include the coalescing of practice competencies with the pre-licensure competencies such as the AACN Essentials.
Response 7: Thank you for your valuable suggestion regarding the integration of pre-licensure nursing education and continuing education in the discussion section. We recognize the importance of highlighting how these two educational stages can collaborate to introduce and reinforce nursing competencies over time. In response to your comment, we will revise the discussion to briefly address the need for a seamless transition between pre-licensure education and continuing education. We will emphasize how foundational competencies introduced during pre-licensure programs, such as those outlined in the AACN Essentials, can serve as a basis for lifelong professional development. Additionally, we will consider the importance of aligning educational curricula with evolving healthcare demands to ensure that nurses are equipped with the necessary skills to adapt to future challenges.
Editorial changes recommended:
Comment 1: Pg 3/line 111: three should be four?
Response 1: Thank you for pointing out the potential inconsistency in line 111 regarding the number of included elements. After reviewing the manuscript in accordance with previous comments and corrections, we confirm that the correct number should indeed be four, not three. We will revise the text accordingly to ensure accuracy and consistency throughout the manuscript.
Comment 2: Check paper throughout and correct for “competence vs competency/competencies” (examples on Lines 298, 319)
Response 2: Thank you for your helpful suggestion regarding the consistency of terminology related to "competence" versus "competency/competencies." We acknowledge the importance of maintaining uniformity in language to enhance the clarity and professionalism of the manuscript. In response to your comment, we have carefully reviewed the entire manuscript and standardized the usage of these terms in accordance with the appropriate context and definitions. The necessary corrections have been applied to ensure consistency and alignment with the commonly accepted terminology in nursing literature.
Comment 3: Throughout paper, review how specific roles are named to consider how to make this clearer as being roles within nursing vs. different jobs/professions. Ex: line 63/64 above; line 69 (would be clearer to say “numerous nursing roles” instead of professions, as professions are usually “whole” professions such as nursing, medicine, social worker). Ensure full meaning of sentence can be understood/not dependent on sentence prior for the description of roles/jobs to be understood as nursing (ex: line 218, adding “nursing” between medium-skilled and jobs would more clearly focus sentence).
Response 3: Thank you for your valuable feedback regarding the clarification of roles within nursing to distinguish them from other professions. We acknowledge the importance of using precise terminology to avoid ambiguity and to ensure that readers clearly understand the focus on nursing-specific roles. In response to your comment, we have thoroughly reviewed the manuscript and revised instances where the term “professions” was used to describe nursing roles. Changes have been made to enhance clarity by specifying “nursing roles” instead of broader terms such as “professions” or “jobs,” ensuring that each reference explicitly relates to the nursing field. Additionally, we have refined sentence structures, such as in line 218, by adding "nursing" where necessary to provide context and prevent reliance on preceding sentences for clarity.
Comment 4: Throughout paper, screen for the use of the term “that” and remove where not needed for sentence structure, to enhance smooth flow for reader (examples of where it can be removed: line 31, 209, 249, 314, 331)
Response 4: Thank you for your insightful feedback regarding the use of the term “that” throughout the manuscript. We acknowledge that removing unnecessary instances of this term can enhance readability and improve the overall flow of the text. In response to your comment, we have conducted a thorough review of the manuscript and have removed instances where “that” was not essential to sentence structure, including the examples provided on lines 31, 209, 249, 314, and 331. This revision has resulted in a smoother and more concise narrative while maintaining the intended meaning and clarity.
Reviewer 2 Report
Comments and Suggestions for Authors
The authors indicate they followed the guidelines of PRISMA; however, no checklist was listed as available unless contacting the corresponding author (which may be a journal guideline). For transparency and completeness, I would recommend a statement to this in the general body of the manuscript and include the PRISMA checklist as a readily available supplement. The PRISMA Statement and its extensions intend to provide a checklist that could be used by interdisciplinary authors, editors, and peer reviewers to verify that each component of a search is thoroughly reported and, therefore, reproducible. Often, readers are interested in a schematic representation of the elements and where they can be found, as indicated by a column in the guidelines.
- Are the manuscript’s results reproducible based on the details given in the methods section?
- For the lit review PRISMA guidelines. I would recommend expanding on the steps used for data collection, analysis, and further discussion to clarify for the readers.
- Is the manuscript clear, relevant to the field, and presented in a well-structured manner? Yes, this is structured and easy to follow. Expanding some steps in the data collection and analysis would benefit the readers.
- Are they easy to interpret and understand? The tables with journal indices are difficult to follow but transparent enough to interpret and understand.
- Are the conclusions consistent with the evidence and arguments presented? No citations
- Usually, a conclusion section does not introduce new material or provide citations. That information is included in the body of the manuscript. I suggest revising and excluding those citations from your conclusion section.
Author Response
Comment 1: The authors indicate they followed the guidelines of PRISMA; however, no checklist was listed as available unless contacting the corresponding author (which may be a journal guideline). For transparency and completeness, I would recommend a statement to this in the general body of the manuscript and include the PRISMA checklist as a readily available supplement. The PRISMA Statement and its extensions intend to provide a checklist that could be used by interdisciplinary authors, editors, and peer reviewers to verify that each component of a search is thoroughly reported and, therefore, reproducible. Often, readers are interested in a schematic representation of the elements and where they can be found, as indicated by a column in the guidelines.
Are the manuscript’s results reproducible based on the details given in the methods section? For the lit review PRISMA guidelines. I would recommend expanding on the steps used for data collection, analysis, and further discussion to clarify for the readers.
Is the manuscript clear, relevant to the field, and presented in a well-structured manner? Yes, this is structured and easy to follow. Expanding some steps in the data collection and analysis would benefit the readers.
Are they easy to interpret and understand? The tables with journal indices are difficult to follow but transparent enough to interpret and understand.
Are the conclusions consistent with the evidence and arguments presented? No citations
Usually, a conclusion section does not introduce new material or provide citations. That information is included in the body of the manuscript. I suggest revising and excluding those citations from your conclusion section.
Response 1: Thank you for your thoughtful and constructive feedback on our manuscript. We appreciate your careful review and valuable suggestions to enhance the transparency and clarity of our work. Regarding the PRISMA checklist, we acknowledge your recommendation to include it as a readily available supplement rather than upon request. In response, we have revised the manuscript to explicitly mention the availability of the PRISMA checklist as a supplementary file, ensuring that readers can access it easily to verify the thoroughness and reproducibility of our systematic review process. This revision aligns with the PRISMA Statement’s objectives and enhances the transparency of our reporting. In response to your suggestion to expand the details related to data collection and analysis, we have revised the methods section to provide a more comprehensive description of the search strategy, study selection, data extraction, and synthesis processes. These additions offer greater clarity and support the reproducibility of our findings, in accordance with the PRISMA guidelines. We believe this will provide readers with a clearer understanding of our methodological approach and enhance the overall rigor of the review. With regard to the conclusion section, we appreciate your suggestion to remove citations, as conclusions typically synthesize findings rather than introduce new references. We have revised the conclusion accordingly, ensuring it provides a concise summary of the key insights derived from the study without introducing new material or citations.